# Learning the Learning Rate for Prediction with Expert Advice

**Wouter M. Koolen**
Queensland University of Technology and UC Berkeley
wouter.koolen@qut.edu.au

**Tim van Erven**
Leiden University, the Netherlands
tim@timvanerven.nl

**Peter D. Grünwald**
Leiden University and Centrum Wiskunde & Informatica, the Netherlands
pdg@cwi.nl

## Abstract

Most standard algorithms for prediction with expert advice depend on a parameter called the learning rate. This learning rate needs to be large enough to fit the data well, but small enough to prevent overfitting. For the exponential weights algorithm, a sequence of prior work has established theoretical guarantees for higher and higher data-dependent tunings of the learning rate, which allow for increasingly aggressive learning. But in practice such theoretical tunings often still perform worse (as measured by their regret) than ad hoc tuning with an even higher learning rate. To close the gap between theory and practice we introduce an approach to learn the learning rate. Up to a factor that is at most (poly)logarithmic in the number of experts and the inverse of the learning rate, our method performs as well as if we would know the empirically best learning rate from a large range that includes both conservative small values and values that are much higher than those for which formal guarantees were previously available. Our method employs a grid of learning rates, yet runs in linear time regardless of the size of the grid.

## 1   Introduction

Consider a learner who in each round $t = 1, 2, \ldots$ specifies a probability distribution $\boldsymbol{w}_t$ on $K$ experts, before being told a vector $\boldsymbol{\ell}_t \in [0,1]^K$ with their losses and consequently incurring loss $h_t := \boldsymbol{w}_t \cdot \boldsymbol{\ell}_t$. Losses are summed up over trials and after $T$ rounds the learner's cumulative loss $H_T = \sum_{t=1}^T h_t$ is compared to the cumulative losses $L_T^k = \sum_{t=1}^T \ell_t^k$ of the experts $k = 1, \ldots, K$. This is essentially the framework of *prediction with expert advice* [1, 2], in particular the standard *Hedge setting* [3]. Ideally, the learner's predictions would not be much worse than those of the best expert, who has cumulative loss $L_T^* = \min_k L_T^k$, so that the *regret* $\mathcal{R}_T = H_T - L_T^*$ is small.

*Follow-the-Leader* (FTL) is a natural strategy for the learner. In any round $t$, it predicts with a point mass on the expert $k$ with minimum loss $L_{t-1}^k$, i.e. the expert that was best on the previous $t - 1$ rounds. However, in the standard game-theoretic analysis, the experts' losses are assumed to be generated by an adversary, and then the regret for FTL can grow linearly in $T$ [4], which means that it is not learning. To do better, the predictions need to be less outspoken, which can be accomplished by replacing FTL's choice of the expert with minimal cumulative loss by the soft minimum $w_t^k \propto e^{-\eta L_{t-1}^k}$, which is known as the *exponential weights* or *Hedge* algorithm [3]. Here $\eta > 0$ is a regularisation parameter that is called the *learning rate*. As $\eta \to \infty$ the soft minimum approaches the exact minimum and exponential weights converges to FTL. In contrast, the lower $\eta$, the more the soft minimum resembles a uniform distribution and the more conservative the learner.

Let $\mathcal{R}_T^\eta$ denote the regret for exponential weights with learning rate $\eta$. To obtain guarantees against adversarial losses, several tunings of $\eta$ have been proposed in the literature. Most of these may be understood by starting with the bound

$$\mathcal{R}_T^\eta \le \frac{\ln K}{\eta} + \sum_{t=1}^{T} \delta_t^\eta, \tag{1}$$

which holds for any sequence of losses. Here $\delta_t^\eta \ge 0$ is the approximation error (called *mixability gap* by [5]) when the loss of the learner in round $t$ is approximated by the so-called *mix loss*, which is a certain $\eta$-exp-concave lower bound (see Section 2.1). The analysis then proceeds by giving an upper bound $b_t(\eta) \ge \delta_t^\eta$ and choosing $\eta$ to balance the two terms $\ln(K)/\eta$ and $\sum_t b_t(\eta)$. In particular, the bound $\delta_t^\eta \le \eta/8$ results in the most conservative tuning $\eta = \sqrt{8\ln(K)/T}$, for which the regret is always bounded by $O(\sqrt{T\ln(K)})$; the same guarantee can still be achieved even if the horizon $T$ is unknown in advance by using, for instance, the so-called doubling trick [4]. It is possible though to learn more aggressively by using a bound on $\delta_t^\eta$ that depends on the data. The first such improvement can be obtained by using $\delta_t^\eta \le e^\eta \boldsymbol{w}_t \cdot \boldsymbol{\ell}_t$ and choosing $\eta = \ln(1 + \sqrt{2\ln(K)/L_T^*}) \approx \sqrt{2\ln(K)/L_T^*}$, where again the doubling trick can be used if $L_T^*$ is unknown in advance, which leads to a bound of $O(\sqrt{L_T^*\ln(K)} + \ln K)$ [6, 4]. Since $L_T^* \le T$ this is never worse than the conservative tuning, and it can be better if the best expert has very small losses (a case sometimes called the "low noise condition"). A further improvement has been proposed by Cesa-Bianchi *et al.* [7], who bound $\delta_t^\eta$ by a constant times the variance $v_t^\eta$ of $\ell_t^k$ when $k$ is distributed according to $\boldsymbol{w}_t$, such that $v_t^\eta = \boldsymbol{w}_t \cdot (\boldsymbol{\ell}_t - h_t)^2$. Rather than using a constant learning rate, at time $t$ they play the Hedge weights $\boldsymbol{w}_t$ based on a time-varying learning rate $\eta_t$ that is approximately tuned as $\sqrt{\ln(K)/V_{t-1}}$ with $V_t = \sum_{s\le t} v_s^{\eta_s}$. This leads to a so-called *second-order* bound on the regret of the form

$$\mathcal{R}_T = O\left(\sqrt{V_t \ln(K)} + \ln K\right), \tag{2}$$

which, as Cesa-Bianchi *et al.* show, implies

$$\mathcal{R}_T = O\left(\sqrt{\frac{L_T^*(T - L_T^*)}{T}\ln(K)} + \ln K\right) \tag{3}$$

and is therefore always better than the tuning in terms of $L_T^*$ (note though that (2) can be much stronger than (3) on data for which the exponential weights rapidly concentrate on a single expert, see also [8]). The general pattern that emerges is that the better the bound on $\delta_t^\eta$, the higher $\eta$ can be chosen and the more aggressive the learning. De Rooij *et al.* [5] take this approach to its extreme and do not bound $\delta_t^\eta$ at all. In their *AdaHedge* algorithm they tune $\eta_t = \ln(K)/\Delta_{t-1}$ where $\Delta_t = \sum_{s\le t} \delta_s^{\eta_s}$, which is very similar to the second-order tuning of Cesa-Bianchi *et al.* and indeed also satisfies (2) and (3). Thus, this sequence of prior works appears to have reached the limit of what is possible based on improving the bound on $\delta_t^\eta$. Unfortunately, however, if the data are not adversarial, then even second-order bounds do not guarantee the best possible tuning of $\eta$ for the data at hand. (See the experiments that study the influence of $\eta$ in [5].) In practice, selecting $\eta_t$ to be the best-performing learning rate so far (that is, running FTL at the meta-level) appears to work well [9], but this approach requires a computationally intensive grid search over learning rates [9] and formal guarantees can only be given for independent and identically distributed (IID) data [10]. A new technique based on speculatively trying out different $\eta$ was therefore introduced in the *FlipFlop* algorithm [5]. By alternating learning rates $\eta_t = \infty$ and $\eta_t$ that are very similar to those of AdaHedge, FlipFlop is both able to satisfy the second-order bounds (2) and (3), and to guarantee that its regret is never much worse than the regret $\mathcal{R}_T^\infty$ for Follow-the-Leader:

$$\mathcal{R}_T = O(\mathcal{R}_T^\infty). \tag{4}$$

Thus FlipFlop covers two extremes: on the one hand it is able to compete with $\eta$ that are small enough to deal with the worst case, and on the other hand it can compete with $\eta = \infty$ (FTL).

**Main Contribution** We generalise the FlipFlop approach to cover a large range of $\eta$ in between. As before, let $\mathcal{R}_T^\eta$ denote the regret of exponential weights with fixed learning rate $\eta$. We introduce

the *learning the learning rate* (LLR) algorithm, which satisfies (2), (3) and (4) and in addition guarantees a regret satisfying

$$\mathcal{R}_T = O\left(\ln(K)\left(\ln\tfrac{1}{\eta}\right)^{1+\varepsilon}\mathcal{R}_T^{\eta}\right) \qquad \text{for all } \eta \in [\eta_{t*}^{\mathrm{ah}}, 1] \tag{5}$$

for any $\varepsilon > 0$. Thus, LLR performs almost as well as the learning rate $\hat{\eta}_T \in [\eta_{t*}^{\mathrm{ah}}, 1]$ that is optimal with hindsight. Here the lower end-point $\eta_{t*}^{\mathrm{ah}} \geq (1 - o(1))\sqrt{\ln(K)/T}$ (as follows from (28) below) is a data-dependent value that is sufficiently conservative (i.e. small) to provide second-order guarantees and consequently worst-case optimality. The upper end-point 1 is an artefact of the analysis, which we introduce because, for general losses in $[0, 1]^K$, we do not have a guarantee in terms of $\mathcal{R}_T^{\eta}$ for $1 < \eta < \infty$. For the special case of binary losses $\ell_t \in \{0, 1\}^K$, however, we can say a bit more: as shown in Appendix B of the supplementary material, in this special case the LLR algorithm guarantees regret bounded by $\mathcal{R}_T = O(K\mathcal{R}_T^{\eta})$ for all $\eta \in [1, \infty]$.

The additional factor $\ln(K)\ln^{1+\varepsilon}(1/\eta)$ in (5) comes from a prior on an exponentially spaced grid of $\eta$. It is logarithmic in the number of experts $K$, and its dependence on $1/\eta$ grows slower than $\ln^{1+\varepsilon}(1/\eta) \leq \ln^{1+\varepsilon}(1/\eta_{t*}^{\mathrm{ah}}) = O(\ln^{1+\varepsilon}(T))$ for any $\varepsilon > 0$. For the optimally tuned $\hat{\eta}_T$, we have in mind regret that grows like $\mathcal{R}_T^{\hat{\eta}_T} = O(T^{\alpha})$ for some $\alpha \in [0, 1/2]$, so an additional polylog factor seems a small price to pay to adapt to the right exponent $\alpha$.

Although $\eta \geq \eta_{t*}^{\mathrm{ah}}$ appear to be most important, the regret for LLR can also be related to $\mathcal{R}_T^{\eta}$ for lower $\eta$:

$$\mathcal{R}_T = O\left(\frac{\ln K}{\eta}\right) \qquad \text{for all } \eta < \eta_{t*}^{\mathrm{ah}}, \tag{6}$$

which is not in terms of $\mathcal{R}_T^{\eta}$, but still improves on the standard bound (1) because $\delta_t^{\eta} \geq 0$ for all $\eta$.

The LLR algorithm takes two parameters, which determine the trade-off between constants in the bounds (2)–(6) above. Normally we would propose to set these parameters to moderate values, but if we do let them approach various limits, LLR becomes essentially the same as FlipFlop, AdaHedge or FTL (see Section 2).

We emphasise that we do not just have a bound on LLR that is unavailable for earlier methods; there also exist actual losses for which the optimal learning rate with hindsight $\hat{\eta}_T$ is fundamentally in between the robust learning rates chosen by AdaHedge and the aggressive choice $\eta = \infty$ of FTL. On such data, Hedge with fixed learning rate $\hat{\eta}_T$ performs significantly better than both these extremes; see Figure 1. In Appendix A we describe the data used to generate Figure 1 and explain why the regret obtained by LLR is significantly smaller than the regret of AdaHedge, FTL and all other tunings described above.

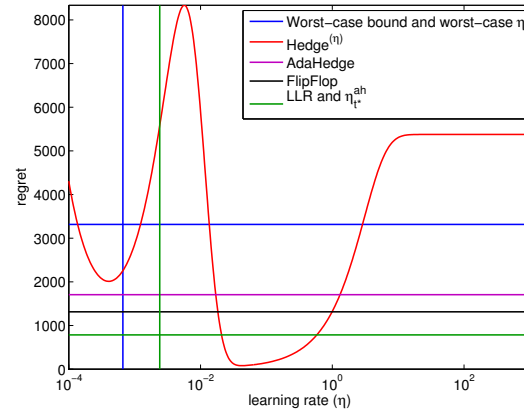

Figure 1: Example data (details in Appendix A) on which Hedge/exponential weights with intermediate learning rate (global minimum) performs much better than both the worst-case optimal learning rate (local minimum on the left) and large learning rates (plateau on the right). We also show the performance of the algorithms mentioned in the introduction.

**Computational Efficiency** Although LLR employs a grid of $\eta$, it does not have to search over this grid. Instead, in each time step it only has to do computations for the single $\eta$ that is active, and, as a consequence, it runs as fast as using exponential weights with a single fixed $\eta$, which is linear in $K$ and $T$. LLR, as presented here, does store information about all the grid points, which requires $O(\ln(K)\ln(T))$ storage, but we describe a simple approximation that runs equally fast and only requires a constant amount of storage.

**Outline** The paper is organized as follows. In Section 2 we define the LLR algorithm and in Section 3 we make precise how it satisfies (2), (3), (4), (5) and (6). Section 4 provides a discussion. Finally, the appendix contains a description of the data in Figure 1 and most of the proofs.

## 2 The Learning the Learning Rate Algorithm

In this section we describe the LLR algorithm, which is a particular strategy for choosing a time-varying learning rate in exponential weights. We start by formally describing the setting and then explain how LLR chooses its learning rates.

### 2.1 The Hedge Setting

At the start of each round $t = 1, 2, \ldots$ the learner produces a probability distribution $\boldsymbol{w}_t = (w_t^1, \ldots, w_t^K)$ on $K \geq 2$ experts. Then the experts incur losses $\boldsymbol{\ell}_t = (\ell_t^1, \ldots, \ell_t^K) \in [0, 1]^K$ and the learner's loss $h_t = \boldsymbol{w}_t \cdot \boldsymbol{\ell}_t = \sum_k w_t^k \ell_t^k$ is the expected loss under $\boldsymbol{w}_t$. After $T$ rounds, the learner's cumulative loss is $H_T = \sum_{t=1}^T h_t$ and the cumulative losses for the experts are $L_T^k = \sum_{t=1}^T \ell_t^k$. The goal is to minimize the regret $\mathcal{R}_T = H_T - L_T^*$ with respect to the cumulative loss $L_T^* = \min_k L_T^k$ of the best expert. We consider strategies for the learner that play the exponential weights distribution

$$w_t^k = \frac{e^{-\eta_t L_{t-1}^k}}{\sum_{j=1}^K e^{-\eta_t L_{t-1}^j}}$$

for a choice of learning rate $\eta_t$ that may depend on all losses before time $t$. To analyse such methods, it is common to approximate the learner's loss $h_t$ by the *mix loss* $m_t = -\frac{1}{\eta_t} \ln \sum_k w_t^k e^{-\eta_t \ell_t^k}$, which appears under a variety of names in e.g. [7, 4, 11, 5]. The resulting approximation error or *mixability gap* $\delta_t = h_t - m_t$ is always non-negative and cannot exceed 1. This, and some other basic properties of the mix loss are listed in Lemma 1 of De Rooij *et al.* [5], which we reproduce as Lemma C.1 in the additional material.

As will be explained in the next section, LLR does not monitor the regrets of all learning rates directly. Instead, it tracks their cumulative mixability gaps, which provide a convenient lower bound on the regret that is monotonically increasing with the number of rounds $T$, in contrast to the regret itself. To show this, let $\mathcal{R}_T^\eta$ denote the regret of the exponential weights strategy with fixed learning rate $\eta_t = \eta$, and similarly let $M_T^\eta = \sum_{t=1}^T m_t^\eta$ and $\Delta_T^\eta = \sum_{t=1}^T \delta_t^\eta$ denote its cumulative mix loss and mixability gap.

**Lemma 2.1.** *For any fixed learning rate $\eta \in (0, \infty]$, the regret of exponential weights satisfies*

$$\mathcal{R}_T^\eta \geq \Delta_T^\eta. \tag{7}$$

*Proof.* Apply property 3 in Lemma C.1 to the regret decomposition $\mathcal{R}_T^\eta = M_T^\eta - L_T^* + \Delta_T^\eta$. $\qquad\square$

We will use the following notational conventions. Lower-case letters indicate instantaneous quantities like $m_t$, $\delta_t$ and $\boldsymbol{w}_t$, whereas uppercase letters denote cumulative quantities like $M_T$, $\Delta_T$ and $\mathcal{R}_T$. In the absence of a superscript the learning rates present in any such quantity are those chosen by LLR. In contrast, the superscript $^\eta$ refers to using the same fixed learning rate $\eta$ throughout.

### 2.2 LLR's Choice of Learning Rate

The LLR algorithm is a member of the exponential weights family of algorithms. Its defining property is its adaptive and non-monotonic selection of the learning rate $\eta_t$, which is specified in Algorithm 1 and explained next. The LLR algorithm works in regimes in which it speculatively tries out different strategies for $\eta_t$. Almost all of these strategies consist of choosing a fixed $\eta$ from the following *grid*:

$$\eta^1 = \infty, \qquad \eta^i = \alpha^{2-i} \quad \text{for } i = 2, 3, \ldots, \tag{8}$$

where the exponential base

$$\alpha = 1 + 1/\log_2 K \tag{9}$$

**Algorithm 1** $\text{LLR}(\pi^{\text{ah}}, \pi^{\infty})$. The grid $\eta^1, \eta^2, \ldots$ and weights $\pi^1, \pi^2, \ldots$ are defined in (8) and (12).

---

Initialise $b_0 := 0$; $\Delta_0^{\text{ah}} := 0$; $\Delta_0^i := 0$ for all $i \geq 1$.
**for** $t = 1, 2, \ldots$ **do**
    **if** all active indices and ah are $b_{t-1}$-full **then**
        Increase $b_t := \phi \Delta_{t-1}^{\text{ah}} / \pi^{\text{ah}}$ (with $\phi$ as defined in (14))
    **else**
        Keep $b_t := b_{t-1}$
    **end if**
    Let $i$ be the least non-$b_t$-full index.
    **if** $i$ is active **then**
        Play $\eta^i$.
        Update $\Delta_t^i := \Delta_{t-1}^i + \delta_t^i$. Keep $\Delta_t^j := \Delta_{t-1}^j$ for $j \neq i$ and $\Delta_t^{\text{ah}} := \Delta_{t-1}^{\text{ah}}$.
    **else**
        Play $\eta_t^{\text{ah}}$ as defined in (10).
        Update $\Delta_t^{\text{ah}} := \Delta_{t-1}^{\text{ah}} + \delta_t^{\text{ah}}$. Keep $\Delta_t^j := \Delta_{t-1}^j$ for all $j \geq 1$.
    **end if**
**end for**

---

is chosen to ensure that the grid is dense enough so that $\eta^i$ for $i \geq 2$ is representative for all $\eta \in [\eta^{i+1}, \eta^i]$ (this is made precise in Lemma 3.3). We also include the special value $\eta^1 = \infty$, because it corresponds to FTL, which works well for IID data and data with a small number of leader changes, as discussed by De Rooij *et al.* [5].

For each *index* $i = 1, 2, \ldots$ in the grid, let $\mathcal{A}_t^i \subseteq \{1, \ldots, t\}$ denote the set of rounds up to trial $t$ in which the LLR algorithm plays $\eta^i$. Then LLR keeps track of the performance of $\eta^i$ by storing the sum of mixability gaps $\delta_t^i \equiv \delta_t^{\eta^i}$ for which $\eta^i$ is responsible:

$$\Delta_t^i = \sum_{s \in \mathcal{A}_t^i} \delta_s^i.$$

In addition to the grid in (8), LLR considers one more strategy, which we will call the *AdaHedge* strategy, because it is very similar to the learning rate chosen by the AdaHedge algorithm [5]. In the AdaHedge strategy, LLR plays $\eta_t$ equal to

$$\eta_t^{\text{ah}} = \frac{\ln K}{\Delta_{t-1}^{\text{ah}}}, \tag{10}$$

where $\Delta_t^{\text{ah}} = \sum_{s \in \mathcal{A}_t^{\text{ah}}} \delta_s^{\text{ah}}$ is the sum of mixability gaps $\delta_t^{\text{ah}} \equiv \delta_t^{\eta_t^{\text{ah}}}$ during the rounds $\mathcal{A}_t^{\text{ah}} \subseteq \{1, \ldots, t\}$ in which LLR plays the AdaHedge strategy. The only difference to the original Ada-Hedge is that the latter sums the mixability gaps over all $s \in \{1, \ldots, t\}$, not just those in $\mathcal{A}_t^{\text{ah}}$. Note that, in our variation, $\eta_t^{\text{ah}}$ does not change during rounds outside $\mathcal{A}_t^{\text{ah}}$.

The AdaHedge learning rate $\eta_t^{\text{ah}}$ is non-increasing with $t$, and (as we will show in Theorem 3.6 below) it is small enough to guarantee the worst-case bound (3), which is optimal for adversarial data. We therefore focus on $\eta > \eta_t^{\text{ah}}$ and call an index $i$ in the grid *active* in round $t$ if $\eta^i > \eta_t^{\text{ah}}$. Let $i_{\max} \equiv i_{\max}(t)$ be the number of grid indices that are active at time $t$, such that $\eta^{i_{\max}(t)} \approx \eta_t^{\text{ah}}$. Then LLR cyclically alternates grid learning rates and the AdaHedge learning rate, in a way that approximately maintains

$$\frac{\Delta_t^1}{\pi^1} \approx \frac{\Delta_t^2}{\pi^2} \approx \ldots \approx \frac{\Delta_t^{i_{\max}}}{\pi^{i_{\max}}} \approx \frac{\Delta_t^{\text{ah}}}{\pi^{\text{ah}}} \qquad \text{for all } t, \tag{11}$$

where $\pi^{\text{ah}} > 0$ and $\pi^1, \pi^2, \ldots > 0$ are fixed weights that control the relative importance of Ada-Hedge and the grid points (higher weight = more important). The LLR algorithm takes as parameters $\pi^{\text{ah}}$ and $\pi^{\infty}$, where $\pi^{\text{ah}}$ only has to be positive, but $\pi^{\infty}$ is restricted to $(0, 1)$. We then choose

$$\pi^1 = \pi^{\infty}, \qquad \pi^i = (1 - \pi^{\infty})\rho(i - 1) \qquad \text{for } i \geq 2, \tag{12}$$

where $\rho$ is a prior probability distribution on $\{1, 2, \ldots\}$. It follows that $\sum_{i=1}^{\infty} \pi^i = 1$, so that $\pi^i$ may be interpreted as a prior probability mass on grid index $i$. For $\rho$, we require a distribution with very

heavy tails (meaning $\rho(i)$ not much smaller than $\frac{1}{i}$), and we fix the convenient choice

$$\rho(i) = \int_{\frac{i-1}{\ln K}}^{\frac{i}{\ln K}} \frac{1}{(x+e)\ln^2(x+e)}\, \mathrm{d}x = \frac{1}{\ln\left(\frac{i-1}{\ln K}+e\right)} - \frac{1}{\ln\left(\frac{i}{\ln K}+e\right)}. \tag{13}$$

We cannot guarantee that the invariant (11) holds exactly, and our algorithm incurs overhead for changing learning rates, so we do not want to change learning rates too often. LLR therefore uses an exponentially increasing budget $b$ and tries grid indices and the AdaHedge strategy in sequence until they exhaust the budget. To make this precise, we say that an index $i$ is $b$-*full* in round $t$ if $\Delta_{t-1}^i/\pi^i > b$ and similarly that AdaHedge is $b$-full in round $t$ if $\Delta_{t-1}^{\mathrm{ah}}/\pi^{\mathrm{ah}} > b$. Let $b_t$ be the budget at time $t$, which LLR chooses as follows: first it initialises $b_0 = 0$ and then, for $t \geq 1$, it tests whether all active indices and AdaHedge are $b_{t-1}$-full. If this is the case, LLR approximately increases the budget by a factor $\phi > 1$ by setting $b_t = \phi\Delta_{t-1}^{\mathrm{ah}}/\pi^{\mathrm{ah}} > \phi b_{t-1}$, otherwise it just keeps the budget the same: $b_t = b_{t-1}$. In particular, we will fix budget multiplier

$$\phi = 1 + \sqrt{\pi^{\mathrm{ah}}}, \tag{14}$$

which minimises the constants in our bounds. Now if, at time $t$, there exists an active index that is not $b_t$-full, then LLR plays the first such index. And if all active indices are $b_t$-full, LLR plays the AdaHedge strategy, which cannot be $b_t$-full in this case by definition of $b_t$. This guarantees that all ratios $\Delta_T^i/\pi_T^i$ are approximately within a factor $\phi$ of each other for all $i$ that are active at time $t^*$, which we define to be the last time $t \leq T$ that LLR plays AdaHedge:

$$t^* = \max \mathcal{A}_T^{\mathrm{ah}}. \tag{15}$$

Whenever LLR plays AdaHedge it is possible, however, that a new index $i$ becomes active and it then takes a while for this index's cumulative mixability gap $\Delta_T^i$ to also grow up to the budget. Since AdaHedge is not played while the new index is catching up, the ratio guarantee always still holds for all indices that were active at time $t^*$.

### 2.3 Choosing the LLR Parameters

LLR has several existing strategies as sub-cases. For $\pi^{\mathrm{ah}} \to \infty$ it essentially becomes AdaHedge. For $\pi^\infty \to 1$ it becomes FlipFlop. For $\pi^\infty \to 1$ and $\pi^{\mathrm{ah}} \to 0$ it becomes FTL. Intermediate values for $\pi^{\mathrm{ah}}$ and $\pi^\infty$ retain the benefits of these algorithms, but in addition allow LLR to compete with essentially all learning rates ranging from worst-case safe to extremely aggressive.

### 2.4 Run time and storage

LLR, as presented here, runs in constant time per round. This is because, in each round, it only needs to compute the weights and update the corresponding cumulative mixability gap for a single learning rate strategy. If the current strategy exceeds its budget (becomes $b_t$-full), LLR proceeds to the next[1]. The memory requirement is dominated by the storage of $\Delta_t^1, \ldots, \Delta_t^{i_{\max}(t)}$, which, following the discussion below (5), is at most

$$i_{\max}(T) = 2 + \frac{\ln \frac{1}{\eta^{i_{\max}(T)}}}{\ln \alpha} \leq 2 + \log_\alpha \frac{1}{\eta_T^{\mathrm{ah}}} = O(\ln(K)\ln(T)).$$

However, a minor approximation reduces the memory requirement down to a constant: At any point in time the grid strategies considered by LLR split in three. Let us say that $\eta^i$ is played at time $t$. Then all preceding $\eta^j$ for $j \leq i$ are already at (or slightly past) the budget. And all succeeding $\eta^j$ for $i < j \leq i_{\max}$ are still at (or slightly past) the previous budget. So we can approximate their cumulative mixability gaps by simply ignoring these slight overshoots. It then suffices to store only the cumulative mixability gap for the currently advancing $\eta^i$, and the current and previous budget.

# 3 Analysis of the LLR algorithm

In this section we analyse the regret of LLR. We first show that for each loss sequence the regret is bounded in terms of the cumulative mixability gaps $\Delta_T^i$ and $\Delta_T^{\text{ah}}$ incurred by the active learning rates (Lemma 3.1). As LLR keeps the cumulative mixability gaps approximately balanced according to (11), we can then further bound the regret in terms of each of the individual learning rates in the grid (Lemma 3.2). The next step is to deal with learning rates between grid points, by showing that their cumulative mixability gap $\Delta_T^\eta$ relates to $\Delta_T^i$ for the nearest higher grid point $\eta^i \geq \eta$ (Lemma 3.3). In Lemma 3.4 we put all these steps together. As the cumulative mixability gap $\Delta_T^\eta$ does not exceed the regret $\mathcal{R}_T^\eta$ for fixed learning rates (Lemma 2.1), we can then derive the bounds (2) through (6) from the introduction in Theorems 3.5 and 3.6.

We start by showing that the regret of LLR is bounded by the cumulative mixability gaps of the learning rates that it plays. The proof, which appears in Section C.4, is a generalisation of Lemma 12 in [5]. It crucially uses the fact that the lowest learning rate played by LLR is the AdaHedge rate $\eta_t^{\text{ah}}$ which relates to $\Delta_t^{\text{ah}}$.

**Lemma 3.1.** *On any sequence of losses, the regret of the LLR algorithm with parameters $\pi^{\text{ah}} > 0$ and $\pi^\infty \in (0,1)$ is bounded by*

$$\mathcal{R}_T \leq \Big(\frac{\phi}{\phi-1}+2\Big)\Delta_T^{\text{ah}} + \sum_{i=1}^{i_{\max}} \Delta_T^i,$$

*where $i_{\max}$ is the largest $i$ such that $\eta^i$ is active in round $T$ and $\phi$ is defined in (14).*

The LLR budgeting scheme keeps the cumulative mixability gaps from Lemma 3.1 approximately balanced according to (11). The next result, proved in Section C.5, makes this precise.

**Lemma 3.2.** *Fix $t^*$ as in (15). Then for each index $i$ that was active at time $t^*$ and arbitrary $j \neq i$:*

$$\Delta_T^j \leq \phi\left(\frac{\pi^j}{\pi^i}\Delta_T^i + \frac{\pi^j}{\pi^{\text{ah}}}\right) + \min\{1, \eta^j/8\}, \tag{16a}$$

$$\Delta_T^j \leq \phi\frac{\pi^j}{\pi^{\text{ah}}}\Delta_T^{\text{ah}} + \min\{1, \eta^j/8\}, \tag{16b}$$

$$\Delta_T^{\text{ah}} \leq \frac{\pi^{\text{ah}}}{\pi^i}\Delta_T^i + 1. \tag{16c}$$

LLR employs an exponentially spaced grid of learning rates that are evaluated using — and played proportionally to — their cumulative mixability gaps. In the next step (which is restated and proved as Lemma C.7 in the additional material) we show that the mixability gap of a learning rate between grid-points cannot be much smaller than that of its next higher grid neighbour. This establishes in particular that an exponential grid is sufficiently fine.

**Lemma 3.3.** *For $\gamma \geq 1$ and for any sequence of losses with values in $[0,1]$:*

$$\delta_t^{\gamma\eta} \leq \gamma e^{(\gamma-1)(\ln K+\eta)}\delta_t^\eta.$$

The preceding results now allow us to bound the regret of LLR in terms of the cumulative mixability gap of any fixed learning rate (which does not exceed its regret by Lemma 2.1) and in terms of the cumulative mixability gap of AdaHedge (which we will use to establish worst-case optimality).

**Lemma 3.4.** *Suppose the losses take values in $[0,1]$, let $\pi^{\text{ah}} > 0$ and $\pi^\infty \in (0,1)$ be the parameters of the LLR algorithm, and abbreviate $B = \big(\frac{\phi}{\phi-1}+2\big)\pi^{\text{ah}} + \phi$. Then the regret of the LLR algorithm is bounded by*

$$\mathcal{R}_T \leq B\alpha e^{(\alpha-1)(\ln K+1)}\frac{\Delta_T^\eta}{\pi^{i(\eta)}} + \left(\frac{\alpha}{8(\alpha-1)} + \frac{\phi}{\pi^{\text{ah}}} + \frac{\phi}{\phi-1} + 3\right)$$

*for all $\eta \in [\eta_{t^*}^{\text{ah}}, 1]$, where $i(\eta) = 2 + \lfloor \log_\alpha(1/\eta)\rfloor$ is the index of the nearest grid point above $\eta$, and by*

$$\mathcal{R}_T \leq B\frac{\Delta_T^\infty}{\pi^\infty} + \left(\frac{\alpha}{8(\alpha-1)} + \frac{\phi}{\pi^{\text{ah}}} + \frac{\phi}{\phi-1} + 3\right)$$

*for $\eta = \infty$. In addition*

$$\mathcal{R}_T \leq B \frac{\Delta_T^{\text{ah}}}{\pi^{\text{ah}}} + \frac{\alpha}{8(\alpha - 1)} + 1,$$

*and for any $\eta < \eta_{t*}^{\text{ah}}$*

$$\Delta_T^{\text{ah}} \leq \frac{\ln K}{\eta} + 1.$$

The proof appears in additional material Section C.6.

We are now ready for our main result, which is proved in Section C.7. It shows that LLR competes with the regret of any learning rate above the worst-case safe rate and below 1 modulo a mild factor. In addition, LLR also performs well on all data favoured by Follow-the-Leader.

**Theorem 3.5.** *Suppose the losses take values in $[0,1]$, let $\pi^{\text{ah}} > 0$ and $\pi^{\infty} \in (0,1)$ be the parameters of the LLR algorithm, and introduce the constants $B = 1 + 2\sqrt{\pi^{\text{ah}}} + 3\pi^{\text{ah}}$ and $C_K = (\log_2 K + 1)/8 + B/\pi^{\text{ah}} + 1$. Then the regret of LLR is simultaneously bounded by*

$$\mathcal{R}_T \leq \frac{4Be^1}{1 - \pi^{\infty}} (\log_2 K + 1) \underbrace{\ln(7/\eta) \ln^2\big(2\log_2(5/\eta)\big)}_{=O\big(\ln^{1+\varepsilon}(1/\eta)\big) \text{ for any } \varepsilon > 0} \mathcal{R}_T^{\eta} + C_K \qquad \text{for all } \eta \in [\eta_{t*}^{\text{ah}}, 1]$$

*and by*

$$\mathcal{R}_T \leq \frac{B}{\pi^{\infty}} \mathcal{R}_T^{\infty} + C_K \qquad \text{for } \eta = \infty.$$

*In addition*

$$\mathcal{R}_T \leq \frac{B}{\pi^{\text{ah}}} \frac{\ln K}{\eta} + C_K \qquad \text{for any } \eta < \eta_{t*}^{\text{ah}}.$$

To interpret the theorem, we recall from the introduction that $\ln(1/\eta)$ is better than $O(\ln T)$ for all $\eta \geq \eta_{t*}^{\text{ah}}$.

We finally show that LLR is robust to the worst-case. We do this by showing something much stronger, namely that LLR guarantees a so-called second-order bound (a concept introduced in [7]). The bound is phrased in terms of the cumulative variance $V_T = \sum_{t=1}^{T} v_t$, where $v_t = \mathbb{V}_{k \sim \boldsymbol{w}_t}\big[\ell_t^k\big]$ is the variance of $\ell_t^k$ for $k$ distributed according to $\boldsymbol{w}_t$. See Section C.8 for the proof.

**Theorem 3.6.** *Suppose the losses take values in $[0,1]$, let $\pi^{\text{ah}} > 0$ and $\pi^{\infty} \in (0,1)$ be the parameters of the LLR algorithm, and introduce the constants $B = \big(\frac{\phi}{\phi-1} + 2\big)\pi^{\text{ah}} + \phi$ and $C_K = (\log_2 K + 1)/8 + B/\pi^{\text{ah}} + 1$. Then the regret of LLR is bounded by*

$$\mathcal{R}_T \leq \frac{B}{\pi^{\text{ah}}} \sqrt{V_T \ln K} + \left(C_K + \frac{2B \ln K}{3\pi^{\text{ah}}}\right)$$

*and consequently by*

$$\mathcal{R}_T \leq \frac{B}{\pi^{\text{ah}}} \sqrt{\frac{L_T^*(T - L_T^*)}{T} \ln K} + 2\left(C_K + \frac{2B \ln K}{3\pi^{\text{ah}}} + \frac{B^2 \ln K}{(\pi^{\text{ah}})^2}\right).$$

## 4  Discussion

We have shown that our new LLR algorithm is able to recover the same second-order bounds as previous methods, which guard against worst-case data by picking a small learning rate if necessary. What LLR adds is that, at the cost of a (poly)logarithmic overhead factor, it is also able to learn a range of higher learning rates $\eta$, which can potentially achieve much smaller regret (see Figure 1). This is accomplished by covering this range with a grid of sufficient granularity. The overhead factor depends on a prior on the grid, for which we have fixed a particular choice with a heavy tail. However, the algorithm would also work with any other prior, so if it were known a priori that certain values in the grid were of special importance, they could be given larger prior mass. Consequently, a more advanced analysis demonstrating that only a subset of learning rates could potentially be optimal (in the sense of minimizing the regret $\mathcal{R}_T^{\eta}$) would directly lead to factors of improvement in the algorithm. Thus we raise the open question: what is the smallest subset $\mathcal{E}$ of learning rates such that, for any data, the minimum of the regret over this subset $\min_{\eta \in \mathcal{E}} \mathcal{R}_T^{\eta}$ is approximately the same as the minimum $\min_{\eta} \mathcal{R}_T^{\eta}$ over all or a large range of learning rates?

## Footnotes

[1] In the early stages it may happen that the next strategy is already over the budget and needs to be skipped, but this start-up effect quickly disappears when the budget exceeds 1, as the weighted increment $\delta_t^i/\pi^i \leq \eta^i/8\log^{1+\epsilon}(1/\eta)$ is bounded for all $0 \leq \eta \leq 1$.

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
