[Supplementary Material]

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

## A  Simulation Study

Figure 1 shows that an intermediate learning rate $\hat{\eta}_T$ can outperform both the robust small learning rates chosen by methods like AdaHedge and the aggressive large learning $\eta = \infty$ chosen by FTL. In this section, we first discuss the features of Figure 1 in more detail. Then we describe how we generated the underlying data and we explain why the regret obtained by LLR is significantly smaller than the regret of AdaHedge, FTL and the other methods described in the introduction.

### A.1  Interpretation

The red line in Figure 1 shows the regret $\mathcal{R}_T^{\eta}$ of the exponential weights algorithm with a fixed learning rate $\eta_t = \eta$ as a function of $\eta$. Its minimum at $\hat{\eta}_T \approx 1/70$ is the optimal learning rate in hindsight, with corresponding regret $\mathcal{R}_T^{\hat{\eta}_T} \approx 100$. Two blue lines mark the conservative tuning $\eta_T^{\mathrm{wc}} = \sqrt{8(\ln K)/T}$ described in the introduction, and the corresponding worst-case regret bound $\mathcal{R}_T^{\mathrm{wc}} \leq \sqrt{T \ln(K)/2}$. As can be seen, both the bound for $\eta_T^{\mathrm{wc}}$ and its actual regret are substantially larger (about 2200) than the global minimum at $\hat{\eta}_T$. The regret of AdaHedge is indicated by the purple horizontal line, and this line may also be taken as indicative of the performance of the second-order tuning of Cesa-Bianchi *et al.* [7], which is very similar. Although smaller than the regret for the worst-case tuning $\eta_T^{\mathrm{wc}}$, the regret for these second-order methods is still much larger than $\mathcal{R}_T^{\hat{\eta}_T}$. The reason is that these methods use learning rates that are too small. On the other hand, large learning rates (in particular $\eta = \infty$ as used by FTL) also perform much worse than the best possible learning rate, so it is important to find the right intermediate value. This is the objective of LLR (the green line; we used parameters $\pi^{\mathrm{ah}} = \pi^{\infty} = 1/5$), which achieves the smallest regret of all adaptive algorithms described in the introduction. Thus, this data pattern illustrates that intermediate learning rates can be optimal on some data, and motivates the search for an adaptive algorithm like LLR that can learn them. The remaining gap between LLR and the optimal learning rate $\hat{\eta}_T$ is the price we pay for learning the learning rate.

### A.2  Data Generating Process

We now explain the data generating process that was used to generate the data in Figure 1. There are $K = 3$ experts, which each receive $T = 2 \cdot 10^7$ losses. Our focus really is on experts 1 and 2, because the third expert always gets the maximal loss 1; we explain why we include it further below. On a high level, our method to generate the losses for experts 1 and 2 is as follows: there exist some data for which small $\eta$ is much better than large $\eta$, and there also exist data for which large $\eta$ is much better than small $\eta$. We simply alternate these two types of data, which ensures that some intermediate $\eta$ will be the best. In practice, especially when the number of experts is large, there might be other, more complicated interactions between experts that cause intermediate $\eta$ to be best, but our current approach seems to be the simplest illustration of this phenomenon.

More precisely, $T$ losses for the experts are constructed according to Algorithm 2, which depends on parameters $\alpha > \beta$ and $\gamma$, for which we select the values $\alpha = 1/6$, $\beta = 1/14$ and $\gamma = 1/6$. The pattern of losses for experts 1 and 2 is constructed in four phases, which are repeated $T^{\alpha}$ times. Out of these, the crucial parts are Phase 1 and Phase 3, during which the difference in cumulative loss between experts 1 and 2 stays approximately constant, except that it goes up and down by 1 every two rounds, which we call *wiggling*. Phase 1 takes place at a particular cumulative loss difference designed to punish large learning rates and Phase 3 at another designed to punish small learning rates. Phases 2 and 4 simply take care of the transition from Phase 1 to Phase 3 and *vice versa*. For simplicity, we have ignored rounding issues in our rendering of Algorithm 2, which need to be taken care of to make sure that all phases have integer lengths and that at each end of Phase 4 we have $L_t^1 - L_t^2$ *exactly* equal to $1/2$.

**Phase 1: Punish Large Learning Rates**  In Phase 1, which lasts $T^{1/2-\beta}$ rounds every time it is repeated, the difference in cumulative loss between experts 1 and 2 is approximately 0 and every

---

**Algorithm 2** The Data Generating Process

---
Parameters: $T$, $0 < \alpha < 1/2$, $0 < \beta < \alpha$, $0 < \gamma < 1/2$
**for** $t = 1, 2, \ldots, T$ **do**                                                                         ▷ Expert 3 is always bad
    $\ell_t^3 := 1$
**end for**
$\ell_1^1 := 1/2$ ; $\ell_1^2 := 0$ ; $t := 2$                                                          ▷ Tie-breaker
**for** $j := 1, 2, \ldots, T^\alpha$ **do**
    **for** $i := 1, 2, \ldots, T^{1/2-\beta}$ **do**                                       ▷ Phase 1: Wiggles, $|L_t^1 - L_t^2| = 1/2$
        $\ell_t^1 := 0; \ell_t^2 := 1$ ; $\ell_{t+1}^1 := 1$ ; $\ell_t^2 := 0$ ; $t := t + 2$
    **end for**
    **for** $i = 1, 2, \ldots, T^{1/2-\gamma}$ **do**                                       ▷ Phase 2: Expert 2 gets better than 1
        $\ell_t^1 := 1$ ; $\ell_t^2 := 0$ ; $t := t + 1$
    **end for**
    **for** $i = 1, 2, \ldots, T^{1-\alpha} - T^{1/2-\beta} - 2T^{1/2-\gamma}$ **do**    ▷ Phase 3: Wiggles, $L_t^1 - L_t^2 \approx T^{1/2-\gamma}$
        $\ell_t^1 := 1$ ; $\ell_t^2 := 0$ ; $\ell_{t+1}^1 := 0$ ; $\ell_t^2 := 1$ ; $t := t + 2$
    **end for**
    **for** $i = 1, 2, \ldots, T^{1/2-\gamma}$ **do**                                       ▷ Phase 4: Expert 2 gets worse again
        $\ell_t^1 := 0$ ; $\ell_t^2 := 1$ ; $t := t + 1$
    **end for**                                                                           ▷ Now $L_t^1 - L_t^2 = 1/2$ again
**end for**

---

wiggle causes the leader (the expert with smallest cumulative loss) to change two times. As is well-known, each such leader change leads to an additional regret of $1/2$ for FTL. Since the total number of rounds spent in Phase 1 is $T^\alpha T^{1/2-\beta} = T^{1/2+\epsilon}$ for $\epsilon = \alpha - \beta > 0$, this ensures that FTL will incur regret of order strictly larger than $\sqrt{T}$ and hence will not be competitive with the standard worst-case learning rate $\eta_T^{\text{wc}}$, whose regret grows no faster than $O(\sqrt{T})$. Thus, in Phase 1, FTL and similar large learning rates are ruled out.

**Phase 3: Punish Small Learning Rates**  In Phase 3, the difference in cumulative loss $L_t^1 - L_t^2$ between experts 1 and 2 is approximately $T^{1/2-\gamma}$. This distinguishes between small learning rates $\eta \ll T^{-1/2+\gamma}$ for which the exponential weights are not converged on a single expert during Phase 3 and larger learning rates $\eta \gg T^{-1/2+\gamma}$ for which the exponential weights are converged. Convergence of the weights is important, because it may be derived from Lemma C.2 below that the mixability gap can be approximated as

$$\delta_t^\eta \propto \eta v_t^\eta \qquad \text{for } \eta \le 1,$$

where $v_t^\eta$ is the variance of the exponential weights distribution at time $t$, which is approximately $0$ if and only if the weights are converged on a single expert. And the mixability gap to a large extent controls the regret, which (1) and Lemma 2.1 show to be sandwiched by

$$\sum_{t=1}^T \delta_t^\eta \le \mathcal{R}_T^\eta \le \frac{\ln K}{\eta} + \sum_{t=1}^T \delta_t^\eta.$$

Since $\alpha < 1/2$, the fraction of rounds spent in Phase 3 goes to 1 as $T$ tends to infinity, such that

$$\mathcal{R}_T^\eta \ge T\big(1 + o(1)\big)\eta \qquad \text{for } \eta \ll T^{-1/2+\gamma}.$$

This explains the spike in the regret for $\eta$ between $\eta_T^{\text{wc}} \approx T^{-1/2} \approx 10^{-3.7}$ and $T^{-1/2+\gamma} \approx 10^{-2.4}$, and we see in Figure 1 that the spike continues for somewhat larger $\eta$ as well.

If $\eta$ becomes large enough, however, then Phase 3 stops hurting because $v_t^\eta$ will be very small. This can quantified using Lemma 6 of Van Erven *et al.* [8], which bounds the mixability gap by

$$\delta_t^\eta \lesssim \eta(1 - \max_k w_t^k) \le 1 - w_t^2 \qquad \text{for } \eta \le 1.$$

Assuming that the weight of expert 3 is negligible, we have $1 - w_t^2 \approx \exp\big(-\eta T^{1/2-\gamma}\big)$, which is exponentially small in $T$ for $\eta \gg T^{-1/2+\gamma}$. For such $\eta$ the sum of mixability gaps over all repetitions of Phase 3 is therefore bounded by a constant.

This leaves room for a learning rate $\hat{\eta}_T$ that is significantly larger than $T^{-1/2+\gamma}$ (such that it is not hurt by Phase 3), but at the same time is not so large that it suffers from Phase 1. And indeed our experiments confirm that such an intermediate learning rate minimizes the regret. Choosing $\gamma = 0$, we already find an $\hat{\eta}_T$ that beats $\eta_T^{\mathrm{wc}} \propto T^{-1/2}$ and FTL, but AdaHedge (which chooses a learning rate substantially higher than $T^{-1/2}$ when $\gamma = 0$) and hence FlipFlop are then still competitive with all $\eta$. By choosing $\gamma$ slightly above 0, we find that there exists an $\hat{\eta}_T$ that also significantly beats AdaHedge and FlipFlop. As mentioned above, Figure 1 was obtained for $T = 2 \cdot 10^7$, $\alpha = 1/6$, $\beta = 1/14$ and $\gamma = 1/6$.

**The Role of Expert 3** At the final time $T$, the cumulative losses of experts 1 and 2 differ by $1/2$, a constant. Therefore, if we were to leave out expert 3, which always gets maximal loss, it would actually be optimal to use learning rate 0, i.e. not learn anything at all: Hedge would then predict by a uniform mixture of expert 1 and 2 at all $t$, which would give a regret of at most $1/2$. Including expert 3 ensures that this trivial, non-learning version of Hedge does not perform well, for it would put mass $1/3$ at the bad expert 3. Indeed, if we repeat the experiment without this bad expert we end up with a figure that, unlike Figure 1, has no local minimum at $\eta_0 \approx 7 \cdot 10^{-3}$; the red curve is then increasing on $(0, \eta_0)$, while to the right of $\eta_0$ it still behaves just like in Figure 1.

# B For Binary Losses, LLR Is Also Competitive with All $\eta \in [1, \infty]$.

The following result substantially generalizes the first implication in Theorem 18 of [5], who show that, for $K = 2$ experts and losses in $\{0, 1\}$, unbounded (as $T \to \infty$) regret for FTL implies unbounded regret for Hedge with constant learning rate $\eta_t = \eta$. Note that the case with losses in $\{0, 1\}$ corresponds to prediction with expert advice in which the experts always predict with a 0 or 1, the loss is the $0/1$-loss, and the learner is allowed to judge randomized predictions by their expected loss.

**Theorem B.1.** *Fix any $0 < \eta < \infty$ and $K \in \mathbb{N}$. Consider a loss sequence $\boldsymbol{\ell}_1, \boldsymbol{\ell}_2, \dots$ with each $\boldsymbol{\ell}_t \in \{0, 1\}^K$. Then there is a constant $C > 0$, depending on $\eta$, such that for all $T > 0$, $\mathcal{R}_T^{\eta} \geq C \cdot \mathcal{R}_T^{\infty}$. In particular, for $\eta \geq 1$, the inequality holds for $C \geq 1/(2eK)$.*

The theorem shows that, if one is only interested in regret bounds up to constant factors, and the losses of the experts are guaranteed to be in $\{0, 1\}$, then nothing is lost by only considering $\eta = \infty$ (FTL) and all $\eta < \eta_0$ where $\eta_0$ is some fixed constant; the precise constants, hidden in the result, depends on this choice of $\eta_0$. On the other hand, Example 2 of [5] shows that there are cases with losses in $\{0, 1\}$ in which the regret of FTL is bounded, whereas for $\eta = 1$, $\mathcal{R}^{\eta}$ increases linearly (!) in $T$. Hence, including $\eta = \infty$ is essential. This shows that in the special case of $0/1$-valued losses, the LLR algorithm is really competitive with all interesting values of $\eta$ as long as one is only interested in regret optimality up to constant factors.

*Proof.* Let $\hat{\mathcal{K}}_{t-1}$ be the set of leaders at time $t - 1$, i.e. the set of $k \in \{1, \dots, K\}$ that achieve minimum cumulative loss at time $t - 1$. If there is no leader change at time $t$, i.e. if $\hat{\mathcal{K}}_{t-1} = \hat{\mathcal{K}}_t$, then $\ell_{t,k} = \ell_{t,k'}$ for all $k, k' \in \hat{\mathcal{K}}_{t-1}$ and FTL incurs no regret, i.e. $\mathcal{R}_t^{\infty} = \mathcal{R}_{t-1}^{\infty}$. The other possibility is that there is a leader change, at time $t$, i.e. there is a $k \in \hat{\mathcal{K}}_{t-1}$ with $k \notin \hat{\mathcal{K}}_t$. Then there must be an expert $k' \in \hat{\mathcal{K}}_t$ so that $L_{t,k'} < L_{t,k}$ (because $k$ does not lead any more at time $t$) whereas $L_{t-1,k} \leq L_{t-1,k'}$ (because $k$ leads at time $t - 1$). Since $L_{t-1,k}$ and $L_{t-1,k'}$ are integers and $\ell_{t,k}$ and $\ell_{t,k'}$ are both in $\{0, 1\}$, this implies that we must have $\ell_{t,k} = 1$, $\ell_{t,k'} = 0$, $L_{t-1,k} = L_{t-1,k'}$. It follows that $\hat{\mathcal{K}}_t \cap \hat{\mathcal{K}}_{t-1}$ is nonempty, and each $k_0$ in the intersection has $\ell_{t,k_0} = 0$, and each $k_1 \in \hat{\mathcal{K}}_{t-1} \setminus \hat{\mathcal{K}}_t$ has $\ell_{t+1,k_1} = 1$. Setting $K' \geq 1$ equal to the number of experts in $\hat{\mathcal{K}}_{t-1} \setminus \mathcal{K}_t$, it follows that

$$\mathcal{R}_t^{\infty} = \mathcal{R}_{t-1}^{\infty} + \frac{K'}{|\hat{\mathcal{K}}_{t-1}|} \leq \mathcal{R}_{t-1}^{\infty} + 1.$$

Thus, $\mathcal{R}_T^{\infty} \leq \#(\mathrm{lc})$, where $\#(\mathrm{lc})$ denotes the number of leader changes up till time $T$.

Below we prove that for every $t$ with a leader change, $\delta_t^{\eta} \geq C$, where $C$ is a constant, which is at least $1/(2eK)$ if $\eta \geq 1$. Since by Lemma C.1 below, at all other $t'$, $\delta_{t'}^{\eta} \geq 0$, it follows that

$\Delta_T^\eta \geq C \cdot \#(\text{lc}) \geq C\mathcal{R}_T^\infty$, where the final inequality was shown at the end of the previous paragraph. Since $\mathcal{R}_T^\eta \geq \Delta_T^\eta$ by Lemma 2.1, the result then follows.

Thus, it only remains to prove that $\delta_t^\eta \geq C$ with $C$ as above if there is a leader change at time $t$. To see this, note that by Lemma C.2, we have for each $t$, $\delta_t \geq c_\eta v_t$ where $c_\eta = (e^{-\eta} + \eta - 1)/\eta$ is a constant depending on $\eta$ which, by standard calculus, can be seen to be larger than 0 and increasing for all $\eta > 0$. Thus, for $\eta \geq 1$, $c_\eta \geq c_1 = e^{-1}$ and it is sufficient if we can show that, if there is a leader change at time $t$, then $v_t \geq 1/(2K)$. But we know that at time $t-1$, there must be at least two leaders (denoted $k$ and $k'$ above). Since these have maximal weights and weights sum to 1, both of these must have weight at least $1/K$. Using that $\ell_{t,k} = 1$ and $\ell_{t,k'} = 0$, we have

$$v_t = \boldsymbol{w}_t \cdot (\boldsymbol{\ell}_t - h_t)^2 \geq \frac{1}{K}(1 - h_t)^2 + \frac{1}{K}h_t^2 \geq \frac{1}{2K},$$

where we used that $\min_{a \in [0,1]}(1-a)^2 + a^2 = 1/2$. This finishes the proof. $\qquad\square$

## C   Proofs

This section is dedicated to the proofs referenced in the main exposition.

### C.1   Lemma C.1: Basic Properties of the Mix Loss

The following lemma is proved in [5].

**Lemma C.1** (Mix Loss with Constant Learning Rate). *For any learning rate $\eta \in (0, \infty]$*

1. *$0 \leq m_t^\eta \leq h_t^\eta \leq 1$, so that $0 \leq \delta_t^\eta \leq 1$.*

2. *Cumulative mix loss telescopes: $M_T^\eta = \begin{cases} -\frac{1}{\eta}\ln\left(\sum_k w_1^k e^{-\eta L_T^k}\right) & \text{for } \eta < \infty, \\ L_T^* & \text{for } \eta = \infty. \end{cases}$*

3. *Cumulative mix loss approximates the loss of the best expert: $L_T^* \leq M_T^\eta \leq L_T^* + \dfrac{\ln K}{\eta}$.*

4. *The cumulative mix loss $M_T^\eta$ is non-increasing in $\eta$.*

### C.2   Bernstein Sandwich

Here we show that the mixability gap $\delta_t$ is well approximated by the variance $v_t = \boldsymbol{w}_t \cdot (\boldsymbol{\ell}_t - h_t)^2$ for small learning rates $\eta$.

**Lemma C.2** (Bernstein Sandwich). *For $\boldsymbol{\ell}_t \in [0,1]^K$ and $\eta > 0$*

$$\frac{(e^{-\eta} + \eta - 1)}{\eta}v_t \leq \delta_t \leq \frac{(e^\eta - \eta - 1)}{\eta}v_t.$$

*Proof.* As $(e^x - x - 1)/x^2$ is increasing in $x$, all $x \in [-1, 1]$ satisfy

$$e^{-\eta} + \eta - 1 \leq \frac{e^{\eta x} - \eta x - 1}{x^2} \leq e^\eta - \eta - 1.$$

Combination with Lemma C.4 results in

$$(e^{-\eta} + \eta - 1)\min_\lambda \frac{1}{\eta}\sum_k w_k(\lambda - \ell_t^k)^2 \leq \delta_t \leq (e^\eta - \eta - 1)\min_\lambda \frac{1}{\eta}\sum_k w_k(\lambda - \ell_t^k)^2$$

The lemma follows by plugging in the optimizer $\lambda = \boldsymbol{w} \cdot \boldsymbol{\ell}_t$. $\qquad\square$

## C.3 Proof of Lemma 3.3, restated as Lemma C.7

We build up to the proof using a series of lemmas.

**Lemma C.3.** *Let* $w_t^{\eta,k} = \frac{e^{-\eta L_{t-1}^k}}{\sum_j e^{-\eta L_{t-1}^j}}$ *be the exponential weights distribution on* $K$ *experts for learning rate* $\eta > 0$ *and let* $\gamma \geq 1$. *Then*

$$w_t^{\gamma\eta,k} \leq K^{\gamma-1} w_t^{\eta,k} \qquad \text{for all } k. \tag{17}$$

*Proof.* By the log-sum inequality (see [12])

$$
\ln \frac{w_t^{\gamma\eta,k}}{w_t^{\eta,k}} = \ln \frac{\sum_j e^{-\eta(L_{t-1}^j - L_{t-1}^k)}}{\sum_j e^{-\gamma\eta(L_{t-1}^j - L_{t-1}^k)}}
$$

$$
\leq \sum_j \frac{e^{-\eta(L_{t-1}^j - L_{t-1}^k)}}{\sum_j e^{-\eta(L_{t-1}^j - L_{t-1}^k)}} \ln \frac{e^{-\eta(L_{t-1}^j - L_{t-1}^k)}}{e^{-\gamma\eta(L_{t-1}^j - L_{t-1}^k)}}
$$

$$
= (\gamma - 1) \sum_j \frac{e^{-\eta(L_{t-1}^j - L_{t-1}^k)}}{\sum_j e^{-\eta(L_{t-1}^j - L_{t-1}^k)}} \eta (L_{t-1}^j - L_{t-1}^k)
$$

$$
\leq (\gamma - 1)\left( -\sum_j \frac{e^{-\eta(L_{t-1}^j - L_{t-1}^k)}}{\sum_j e^{-\eta(L_{t-1}^j - L_{t-1}^k)}} \ln \left( \frac{e^{-\eta(L_{t-1}^j - L_{t-1}^k)}}{\sum_j e^{-\eta(L_{t-1}^j - L_{t-1}^k)}} \right) \right).
$$

The second inequality follows by $\sum_j e^{-\eta(L_{t-1}^j - L_{t-1}^k)} \geq e^{-\eta(L_{t-1}^k - L_{t-1}^k)} = 1$. Upper bounding that Shannon entropy by $\ln K$ results in (17). $\qquad\square$

**Lemma C.4.** *Fix any learning rate* $\eta$ *and probability vector* $\boldsymbol{w}$. *Let* $\delta_t^\eta(\boldsymbol{w}) = \boldsymbol{w} \cdot \boldsymbol{\ell}_t - m_t^\eta(\boldsymbol{w})$ *be the mixability gap of* $\boldsymbol{w}$, *where* $m_t^\eta(\boldsymbol{w}) = \frac{-1}{\eta} \ln \sum_k w_k e^{-\eta \ell_t^k}$ *is the mix loss of* $\boldsymbol{w}$. *Then*

$$\delta_t^\eta(\boldsymbol{w}) = \min_\lambda \frac{1}{\eta} \sum_k w_k \psi\big(\eta(\lambda - \ell_t^k)\big)$$

*where* $\psi(x) = e^x - x - 1$ *and the minimum is achieved by* $\lambda = m_t^\eta(\boldsymbol{w})$.

*Proof.* Let $\triangle$ be the probability simplex on $K$ outcomes. We will use that, up to scaling, the mix loss is the convex conjugate of the Kullback-Leibler divergence $D(\boldsymbol{v} \| \boldsymbol{w}) = \sum_k v_k \ln \frac{v_k}{w_k}$:

$$-\eta m_t^\eta(\boldsymbol{w}) = \sup_{\boldsymbol{v} \in \triangle} \boldsymbol{v} \cdot (-\eta \boldsymbol{\ell}_t) - D(\boldsymbol{v} \| \boldsymbol{w}).$$

As the Kullback-Leibler may be extended off the simplex to $D(\boldsymbol{v} \| \boldsymbol{w}) = \sum_k (v_k \ln \frac{v_k}{w_k} - v_k + w_k)$ for any vectors $\boldsymbol{v}$ and $\boldsymbol{w}$ with non-negative components, we may introduce a Lagrange multiplier $\lambda$ to enforce the restriction to the simplex and reason as follows:

$$
m_t^\eta(\boldsymbol{w}) = \inf_{\boldsymbol{v} \in \triangle} \frac{1}{\eta} D(\boldsymbol{v} \| \boldsymbol{w}) + \boldsymbol{v} \cdot \boldsymbol{\ell}_t
$$

$$
= \sup_\lambda \inf_{\boldsymbol{v} \in \mathbb{R}_+^K} \frac{1}{\eta} D(\boldsymbol{v} \| \boldsymbol{w}) + \boldsymbol{v} \cdot \boldsymbol{\ell}_t - \lambda(\boldsymbol{1} \cdot \boldsymbol{v} - 1)
$$

$$
= \sup_\lambda \frac{1}{\eta} \sum_k w_k \left( 1 - e^{\eta(\lambda - \ell_t^k)} \right) + \lambda
$$

$$
= \boldsymbol{w} \cdot \boldsymbol{\ell}_t - \inf_\lambda \frac{1}{\eta} \sum_k w_k \psi \big( \eta(\lambda - \ell_t^k) \big),
$$

from which the result follows. $\qquad\square$

**Lemma C.5** (Continuous Log-Sum Inequality). *Let $f, g \colon [a, b] \to \mathbb{R}$ be positive functions such that $\int_a^b g(x)\mathrm{d}x < \infty$. Then*

$$\ln \frac{\int_a^b f(x)\mathrm{d}x}{\int_a^b g(x)\mathrm{d}x} \leq \int_a^b \frac{f(x)}{\int_a^b f(y)\mathrm{d}y} \ln \frac{f(x)}{g(x)} \mathrm{d}x.$$

*Proof.* Let $h(x) = f(x)/g(x)$. Then, by Jensen's inequality and convexity of $z \ln z$,

$$\begin{aligned}
\int_a^b \frac{f(x)}{\int_a^b g(y)\mathrm{d}y} \ln \frac{f(x)}{g(x)} \mathrm{d}x &= \int_a^b \frac{g(x)}{\int_a^b g(y)\mathrm{d}y} \Big( h(x) \ln h(x) \Big) \mathrm{d}x \\
&\geq \Big( \int_a^b \frac{g(x)}{\int_a^b g(y)\mathrm{d}y} h(x)\mathrm{d}x \Big) \ln \Big( \int_a^b \frac{g(x)}{\int_a^b g(y)\mathrm{d}y} h(x)\mathrm{d}x \Big) \\
&= \frac{\int_a^b f(x)\mathrm{d}x}{\int_a^b g(y)\mathrm{d}y} \ln \frac{\int_a^b f(x)\mathrm{d}x}{\int_a^b g(y)\mathrm{d}y}.
\end{aligned}$$

Dividing both sides by $\frac{\int_a^b f(x)\mathrm{d}x}{\int_a^b g(y)\mathrm{d}y}$, the result follows. $\qquad\square$

**Lemma C.6.** *Let $\psi(x) = e^x - x - 1$. Then for $\gamma \geq 1$ and $x \leq B$ for $B \geq 0$*

$$\frac{\psi(\gamma x)}{\psi(x)} \leq \gamma^2 e^{(\gamma-1)B}.$$

*Proof.* We use that

$$\psi(x) = x^2 \int_0^1 (1-u)e^{xu} \, \mathrm{d}u.$$

By the log-sum inequality (c.f. Lemma C.5)

$$\begin{aligned}
\ln \frac{\psi(\gamma x)}{\psi(x)} &\leq \int_0^1 \frac{(\gamma x)^2(1-u)e^{\gamma xu}}{\psi(\gamma x)} \ln \frac{(\gamma x)^2(1-u)e^{\gamma xu}}{x^2(1-u)e^{xu}} \, \mathrm{d}u \\
&= \ln \gamma^2 + (\gamma-1)x \int_0^1 \frac{(\gamma x)^2(1-u)e^{\gamma xu}}{\psi(\gamma x)} u \, \mathrm{d}u \\
&\leq \ln \gamma^2 + (\gamma-1)B \int_0^1 \frac{(\gamma x)^2(1-u)e^{\gamma xu}}{\psi(\gamma x)} u \, \mathrm{d}u \\
&\leq \ln \gamma^2 + (\gamma-1)B,
\end{aligned}$$

where the last inequality uses $u \leq 1$. $\qquad\square$

**Lemma C.7.** *Fix $\eta > 0$ and $\gamma \geq 1$. Let $\boldsymbol{w}^\eta$ be the exponential weight distribution with learning rate $\eta$ (as defined in Lemma C.3) and let $\delta^\eta(\boldsymbol{w})$ be the mixability gap of $\boldsymbol{w}$ as defined in Lemma C.4. Then for any $\boldsymbol{\ell}_t \in [0, 1]^K$*

$$\delta_t^{\gamma\eta}(\boldsymbol{w}^{\gamma\eta}) \leq \gamma e^{(\gamma-1)\eta} K^{1-\gamma} \delta_t^\eta(\boldsymbol{w}^\eta).$$

*Proof.* Substituting the sub-optimal $\lambda = m_t^\eta(\boldsymbol{w}^\eta)$ into the expression for $\delta_t^{\gamma\eta}(\boldsymbol{v})$ given by Lemma C.4, using Lemma C.3 and $\psi \geq 0$, followed by Lemma C.6 we find

$$\begin{aligned}
\delta_t^{\gamma\eta}(\boldsymbol{w}^{\gamma\eta}) &\leq \frac{1}{\gamma\eta} \sum_k w_t^{\gamma\eta,k} \psi\big(\gamma\eta(m_t^\eta(\boldsymbol{w}^\eta) - \ell_t^k)\big) \\
&\leq K^{\gamma-1} \frac{1}{\gamma\eta} \sum_k w_t^{\eta,k} \psi\big(\gamma\eta(m_t^\eta(\boldsymbol{w}^\eta) - \ell_t^k)\big) \\
&\leq K^{\gamma-1} \frac{1}{\gamma\eta} \sum_k w_t^{\eta,k} \psi\big(\eta(m_t^\eta(\boldsymbol{w}^\eta) - \ell_t^k)\big) \gamma^2 e^{(\gamma-1)\eta} \\
&= K^{\gamma-1} e^{(\gamma-1)\eta} \gamma \delta_t^\eta(\boldsymbol{w}^\eta). \qquad\square
\end{aligned}$$

## C.4 Proof of Lemma 3.1

Suppose that after round $T$ LLR has increased its budget $d$ times. For $j = 1, \ldots, d$, let $v_j$ be the last round before the $j$-th increase of the budget, and also define $v_0 = 0$ and $v_{d+1} = T$ for convenience. For $j = 1, \ldots, d+1$, let $M_{[j]} = \sum_{t=v_{j-1}+1}^{v_j} m_t$ be the cumulative mix loss during the $j$-th value of the budget. By construction, the learning rate $\eta_t$ chosen by LLR is non-increasing from round $v_{j-1}+1$ to round $v_j$. Consequently, as the cumulative mix loss $M_t^\eta$ for the first $t$ rounds is non-increasing in $\eta$ (see Lemma C.1),

$$M_{[j]} = m_{v_{j-1}+1} + \sum_{t=v_{j-1}+2}^{v_j} M_t^{\eta_t} - M_{t-1}^{\eta_t} \leq m_{v_{j-1}+1} + \sum_{t=v_{j-1}+2}^{v_j} M_t^{\eta_t} - M_{t-1}^{\eta_{t-1}}$$

$$= m_{v_{j-1}+1} + M_{v_j}^{\eta_{v_j}} - M_{v_{j-1}+1}^{\eta_{v_{j-1}+1}} = M_{v_j}^{\eta_{v_j}} - M_{v_{j-1}}^{\eta_{v_{j-1}+1}} \leq M_{v_j}^{\eta_{v_j}^{\mathrm{ah}}} - M_{v_{j-1}}^{\eta_{v_{j-1}+1}}. \quad (18)$$

For $j = 1$, $M_{v_{j-1}}^{\eta_{v_{j-1}+1}} = M_0^{\eta_{v_{j-1}+1}} = 0 = M_{v_{j-1}}^{\eta_{v_{j-1}}^{\mathrm{ah}}}$; for $j = 2, \ldots, d+1$, property 3 of Lemma C.1 implies

$$M_{v_{j-1}}^{\eta_{v_{j-1}+1}} \geq L_{v_{j-1}}^* \geq M_{v_{j-1}}^{\eta_{v_{j-1}}^{\mathrm{ah}}} - \frac{\ln K}{\eta_{v_{j-1}}^{\mathrm{ah}}} = M_{v_{j-1}}^{\eta_{v_{j-1}}^{\mathrm{ah}}} - \Delta_{v_{j-1}-1}^{\mathrm{ah}} \geq M_{v_{j-1}}^{\eta_{v_{j-1}}^{\mathrm{ah}}} - \Delta_{v_{j-1}}^{\mathrm{ah}}.$$

Combining this with (18), we get

$$M_T = \sum_{j=1}^{d+1} M_{[j]} \leq \sum_{j=1}^{d+1} \left( M_{v_j}^{\eta_{v_j}^{\mathrm{ah}}} - M_{v_{j-1}}^{\eta_{v_{j-1}+1}} \right) \leq \sum_{j=1}^{d+1} \left( M_{v_j}^{\eta_{v_j}^{\mathrm{ah}}} - M_{v_{j-1}}^{\eta_{v_{j-1}}^{\mathrm{ah}}} \right) + \sum_{j=2}^{d+1} \Delta_{v_{j-1}}^{\mathrm{ah}}$$

$$= M_T^{\eta_T^{\mathrm{ah}}} + \sum_{j=1}^{d} \Delta_{v_j}^{\mathrm{ah}} \overset{(\dagger)}{\leq} L_T^* + \Delta_{T-1}^{\mathrm{ah}} + \sum_{j=1}^{d} \Delta_{v_j}^{\mathrm{ah}} \leq L_T^* + \Delta_T^{\mathrm{ah}} + \sum_{j=1}^{d} \Delta_{v_j}^{\mathrm{ah}}.$$

where inequality $(\dagger)$ follows from property 3 of Lemma C.1 and the definition (10) of $\eta_T^{\mathrm{ah}}$. Because the budget has been exceeded $d$ times, we know that

$$\Delta_{v_d}^{\mathrm{ah}} \geq \phi \Delta_{v_{d-1}}^{\mathrm{ah}} \geq \phi^2 \Delta_{v_{d-2}}^{\mathrm{ah}} \geq \ldots \geq \phi^{d-1} \Delta_{v_1}^{\mathrm{ah}},$$

so that

$$\sum_{j=1}^{d} \Delta_{v_j}^{\mathrm{ah}} \leq \sum_{j=1}^{d} \phi^{j-d} \Delta_{v_d}^{\mathrm{ah}} = \Delta_{v_d}^{\mathrm{ah}} \sum_{j=0}^{d-1} \phi^{-j} \leq \Delta_{v_d}^{\mathrm{ah}} \sum_{j=0}^{\infty} \phi^{-j} = \Delta_{v_d}^{\mathrm{ah}} \frac{\phi}{\phi - 1} \leq \Delta_T^{\mathrm{ah}} \frac{\phi}{\phi - 1}.$$

We can now decompose the regret of LLR as

$$\mathcal{R}_T = M_T - L_T^* + \Delta_T \leq \Delta_T^{\mathrm{ah}} + \sum_{j=1}^{d} \Delta_{v_j}^{\mathrm{ah}} + \Delta_T \leq \left( \frac{\phi}{\phi - 1} + 1 \right) \Delta_T^{\mathrm{ah}} + \Delta_T$$

$$= \left( \frac{\phi}{\phi - 1} + 2 \right) \Delta_T^{\mathrm{ah}} + \sum_{i=1}^{i_{\max}} \Delta_T^i,$$

as required.

## C.5 Proof of Lemma 3.2

The value of the budget after $T$ rounds is $b_T$. Assume first that $b_T > 0$. Let $u$ be the round just before the budget was last increased, i.e. $u$ is the last round such that $b_u < b_T$. At time $t^*$, we know $b_{t^*} \geq b_u$ because AdaHedge was played at least once while the budget was $b_u$ to cause its increase. Since $i$ was active at time $t^*$ but AdaHedge was played, $i$ must have been full, i.e. $\Delta_{t^*}^i / \pi^i > b_{t^*}$. Hence

$$b_u \leq b_{t^*} < \Delta_{t^*}^i / \pi^i \leq \Delta_T^i / \pi^i. \quad (19)$$

By definition of the LLR budgeting, $\Delta_t^j/\pi^j \le b_t + \delta_t^j/\pi^j$ and similarly $\Delta_t^{\mathrm{ah}}/\pi^{\mathrm{ah}} \le b_t + \delta_t^{\mathrm{ah}}/\pi^{\mathrm{ah}}$ at any time $t$. By Lemma C.1 $\delta_t^{\tilde\eta} \le 1$, and by Hoeffding's bound on the cumulant generating function [4, Lemma A.1] $\delta_t^\eta \le \eta/8$ regardless of the choice of $\eta$. Hence

$$\frac{\Delta_T^j}{\pi^j} \le b_T + \frac{\min\{1,\eta^j/8\}}{\pi^j} = \phi\Delta_u^{\mathrm{ah}}/\pi^{\mathrm{ah}} + \frac{\min\{1,\eta^j/8\}}{\pi^j}$$

$$\le \phi\left(b_u + \frac{1}{\pi^{\mathrm{ah}}}\right) + \frac{\min\{1,\eta^j/8\}}{\pi^j} \le \phi\left(\frac{\Delta_T^i}{\pi^i} + \frac{1}{\pi^{\mathrm{ah}}}\right) + \frac{\min\{1,\eta^j/8\}}{\pi^j},$$

where the last inequality follows by (19). Similarly, $b_T = \phi\Delta_u^{\mathrm{ah}}/\pi^{\mathrm{ah}} \le \phi\Delta_T^{\mathrm{ah}}/\pi^{\mathrm{ah}}$ implies

$$\frac{\Delta_T^j}{\pi^j} \le b_T + \frac{\min\{1,\eta^j/8\}}{\pi^j} \le \phi\frac{\Delta_T^{\mathrm{ah}}}{\pi^{\mathrm{ah}}} + \frac{\min\{1,\eta^j/8\}}{\pi^j}. \tag{20}$$

For the last bound we use that AdaHedge is played by LLR only after all active $i$ are full (i.e. have exhausted the current budget).

$$\frac{\Delta_T^{\mathrm{ah}}}{\pi^{\mathrm{ah}}} = \frac{\Delta_{t^*}^{\mathrm{ah}}}{\pi^{\mathrm{ah}}} \le b_{t^*} + \frac{\delta_{t^*}^{\mathrm{ah}}}{\pi^{\mathrm{ah}}} < \frac{\Delta_{t^*}^i}{\pi^i} + \frac{\delta_{t^*}^{\mathrm{ah}}}{\pi^{\mathrm{ah}}} \le \frac{\Delta_T^i}{\pi^i} + \frac{1}{\pi^{\mathrm{ah}}}.$$

If $b_T = 0$ then $\eta_{t^*}^{\mathrm{ah}} = \infty$ and hence no $i$ is active at time $t^*$, so we only need to prove (16b). Since $b_T = 0 \le \phi\Delta_T^{\mathrm{ah}}/\pi^{\mathrm{ah}}$ this again follows by (20).

### C.6   Proof of Lemma 3.4

Let $i$ be an arbitrary index that is active at time $t^*$. Then in Lemma 3.1 we bound $\Delta_T^{\mathrm{ah}}$ and $\Delta_T^j$ for $j \ne i$ in terms of $\Delta_T^i$ using Lemma 3.2, which gives

$$\mathcal{R}_T \le \left(\frac{\phi}{\phi-1} + 2\right)\left(\frac{\pi^{\mathrm{ah}}}{\pi^i}\Delta_T^i + 1\right) + \sum_{j \le i_{\max}; j \ne i}\left(\phi\left(\frac{\pi^j}{\pi^i}\Delta_T^i + \frac{\pi^j}{\pi^{\mathrm{ah}}}\right) + \min\{1,\eta^j/8\}\right) + \Delta_T^i$$

$$= \left(\left(\frac{\phi}{\phi-1} + 2\right)\frac{\pi^{\mathrm{ah}}}{\pi^i} + \phi\frac{\sum_{j \le i_{\max}; j \ne i}\pi^j}{\pi^i} + 1\right)\Delta_T^i$$

$$+ \left(\frac{\phi}{\phi-1} + 2 + \phi\frac{\sum_{j \le i_{\max}; j \ne i}\pi^j}{\pi^{\mathrm{ah}}} + \sum_{j \le i_{\max}; j \ne i}\min\{1,\eta^j/8\}\right). \tag{21}$$

By definition $\Delta_T^i$ accumulates $\delta_t^i$ only in rounds $t$ where LLR plays $\eta^i$. Since $\delta_t^i \ge 0$ (see Lemma C.1) we can extend the sum to all trials:

$$\Delta_T^i \le \Delta_T^{\eta^i}. \tag{22}$$

For the sums over $\pi^j$, we have

$$\phi\frac{\sum_{j \le i_{\max}; j \ne i}\pi^j}{\pi^i} + 1 \le \phi\frac{\sum_{j \le i_{\max}}\pi^j}{\pi^i} \le \frac{\phi}{\pi^i} \quad \text{and} \quad \phi\frac{\sum_{j \le i_{\max}; j \ne i}\pi^j}{\pi^{\mathrm{ah}}} \le \frac{\phi}{\pi^{\mathrm{ah}}}. \tag{23}$$

We proceed to bound the sum $\sum_j \min\{1,\eta^j/8\}$, which is at most a constant by the definition of the grid (8). For $j = 1$ the minimum is 1 since $\eta^1 = \infty$, and for $j \ge 2$ we bound the minimum by $\eta^j/8$, which leads to

$$\sum_j \min\{1,\eta^j/8\} \le 1 + \frac{1}{8}\sum_{j=2}^{\infty}\alpha^{2-j} = 1 + \frac{\alpha}{8(\alpha-1)}. \tag{24}$$

Plugging (22), (23) and (24) into (21) for $i = 1$ and using $\pi^1 = \pi^\infty$ (see (12)), we obtain the second inequality of the lemma.

Now let $\eta \in [\eta_{t^*}^{\mathrm{ah}}, 1]$ be arbitrary. Then $i \equiv i(\eta)$ is active at time $t^*$ and $\eta \le \eta^i \le \alpha\eta$, so that by Lemma 3.3 we have

$$\Delta_T^{\eta^i} \le \alpha e^{(\alpha-1)(\ln K + \eta)}\Delta_T^\eta \le \alpha e^{(\alpha-1)(\ln K + 1)}\Delta_T^\eta. \tag{25}$$

Plugging (22), (23), (24) and (25) into (21) for $i = i(\eta)$ establishes the first inequality of the lemma.

Lemma 3.1 combined with (16b) results in

$$\mathcal{R}_T \leq \Big(\frac{\phi}{\phi - 1} + 2\Big)\Delta_T^{\text{ah}} + \phi\frac{\sum_{j=1}^{i_{\max}} \pi^j}{\pi^{\text{ah}}}\Delta_T^{\text{ah}} + \sum_{j \leq i_{\max}} \min\{1, \eta^j/8\},$$

and the third inequality of the theorem follows by $\sum_{i=1}^{i_{\max}} \pi^i \leq 1$ and (24).

Finally, suppose that $\eta < \eta_{t^*}^{\text{ah}}$. Then, since $t^*$ is the last round in which AdaHedge was used and $\delta_{t^*}^{\text{ah}} \leq 1$ by Lemma C.1,

$$\Delta_T^{\text{ah}} = \Delta_{t^*-1}^{\text{ah}} + \delta_{t^*}^{\text{ah}} \leq \Delta_{t^*-1}^{\text{ah}} + 1 = \frac{\ln K}{\eta_{t^*}^{\text{ah}}} + 1 \leq \frac{\ln K}{\eta} + 1,$$

which proves the last inequality of the lemma.

## C.7  Proof of Theorem 3.5

We continue from Lemma 3.4, and start by bounding $1/\pi^{i(\eta)}$ from above. To this end, we first observe that

$$i(\eta) \leq 2 + \log_\alpha(1/\eta) = 2 + \frac{\ln(1/\eta)}{\ln(1 + 1/\log_2 K)} \leq 2 + \big(\log_2 K + 1\big)\ln(1/\eta),$$

where the second inequality follows from $\ln(1 + x) \geq x/(1 + x)$. Next we lower bound the heavy-tailed prior $\rho$. We bound its defining integral (13) by the width times the lowest integrand to find

$$\rho(i) \geq \frac{1}{\ln K \big(\frac{i}{\ln K} + e\big)\ln^2\big(\frac{i}{\ln K} + e\big)}.$$

Hence by the definition of the grid-point weights (12)

$$\frac{1 - \pi^\infty}{\pi^{i(\eta)}} \leq (i(\eta) - 1 + e\ln K)\ln^2\Big(\frac{i(\eta) - 1}{\ln K} + e\Big)$$

$$\leq \big((\log_2 K + 1)\ln(1/\eta) + e\ln K + 1\big)\ln^2\Big(\frac{1 + (\log_2 K + 1)\ln(1/\eta)}{\ln K} + e\Big).$$

The first factor is at most

$$(\log_2 K + 1)\ln(1/\eta) + e\ln K + 1 \leq (\log_2 K + 1)\ln(7/\eta),$$

and we use that $K \geq 2$, so that

$$\frac{1 + (\log_2 K + 1)\ln(1/\eta)}{\ln K} + e \leq \Big(\frac{1}{\ln 2} + \frac{1}{\ln K}\Big)\ln(1/\eta) + \frac{1}{\ln K} + e$$

$$\leq \Big(\frac{1}{\ln 2} + \frac{1}{\ln 2}\Big)\ln(1/\eta) + \frac{1}{\ln 2} + e \leq 2\log_2(5/\eta).$$

Thus

$$\frac{1}{\pi^{i(\eta)}} \leq \frac{\log_2 K + 1}{1 - \pi^\infty}\ln(7/\eta)\ln^2\big(2\log_2(5/\eta)\big). \tag{26}$$

Next, we use the definition of $\alpha$ (9) and $K \geq 2$ to bound

$$\alpha e^{(\alpha-1)(\ln K+1)} = (1 + 1/\log_2 K)e^{\ln 2 + 1/\log_2 K} \leq 4e. \tag{27}$$

The first inequality of the theorem now follows by applying Lemma 2.1, plugging (26), (27) and (7) into Lemma 3.4, and evaluating $\frac{\alpha}{\alpha-1} = \log_2 K + 1$. The second inequality follows directly by plugging in (7). And, finally, the third inequality follows by combining the last two inequalities of Lemma 3.4.

## C.8 Proof of Theorem 3.6

Let $V_T^{\text{ah}} = \sum_{s \in \mathcal{A}_T^{\text{ah}}} v_t$ be the sum of variances $v_t$ over all times $t \leq T$ that LLR plays AdaHedge. By the same argument as in the proofs of Lemma 5 and Theorem 6 in [5]

$$\Delta_T^{\text{ah}} \leq \sqrt{V_T^{\text{ah}} \ln K} + \left(\tfrac{2}{3} \ln K + 1\right). \tag{28}$$

Plugging this bound into the third inequality of Lemma 3.4, bounding $V_T^{\text{ah}} \leq V_T$ and evaluating $\frac{\alpha}{\alpha - 1} = \log_2 K + 1$ and $\phi = 1 + \sqrt{\pi^{\text{ah}}}$, we obtain the first inequality of the theorem. The second inequality follows from the first by the same argument as in the proof of Corollary 3 of [7].