[Reviews · NeurIPS 2014]

Submitted by Assigned_Reviewer_6

This paper presents an algorithm for prediction with expert advice which does not need the learning rate as input parameter. As the theorems in the paper suggest, the algorithm's performance is competitive with the multiplicative weights algorithm given any learning rate, under any adversarial loss sequence. The algorithm follows the multiplicative weights algorithm while chooses the learning rate in every time step based on statistics collected thus far.

The writing is generally fine with some minor language hiccups. The idea of the algorithm for learning the right learning rate is very original. The regret bound proofs seem rather complicated, I could not thoroughly check them. It is also not entirely clear to me if the new parameters that the algorithm needs as input are robust enough so that we do not need to learn them, too.
Summary: Very original algorithmic contribution that makes choosing the learning rate for the exponential weights algorithm obsolete.

Submitted by Assigned_Reviewer_19

The paper proposes a Learning the Learning Rate (LLR) algorithm for learning from expert advice which while retaining several worst case bounds for the problem enjoyed by previous algorithms like exponential weights algorithm, in addition also enjoys a guarantee on regret that is at most a poly logarithmic factor in number of experts and inversely in learning rate to the regret of exponential weights algorithm with that prescribed learning rate (for learning rates in a prescribed interval). Further the proposed algorithm employs a grid over learning rates in the interval and yet runs in linear time regardless of the granularity of the grid. The problem of adaptively learning from expert advice in a way which while retains worst case guarantees also enjoys better practical performance against benign adversaries has been a problem of great interest. The algorithm proposed first of all neatly combines three adaptive bounds from previous works, the variance type bound, the L* type bound for small losses and a more recent bound of the order of follow-the-leader algorithm which has been shown to work well in practice (eg. for iid data).

Overall the paper is well written and the problem considered is of practical interest. Some remarks :

1. I do have some concerns over the lower limit of learning rate used though. Yes, it is true that sqrt{log K / T} is as small a learning rate one needs to consider to be able to guarantee the absolute worst case performance for prediction with experts advice. However in the regret bound you provide, the bound is with respect to eta chosen in hindsight. So it would make sense to provide bounds for even smaller learning rates. I wonder if one could get a guarantee over entire range of step sizes while worsening the regret bound by an additive factor beyond only the multiplicative factor over regret bound of best eta.

2. The paper only considers hind sign bounds w.r.t. to experts algorithms with optimal fixed step sizes. However in practice often step size schemes that vary with time are considered. For instance one might consider step sizes of form eta_t = eta / t^a where eta and a are tuned. The result in this paper only refers to a = 0. At least glancing from the results it is not clear to me that the results can be adapted to the non-zero a case.
Summary: Good paper, would be better if results can be extended to time dependent step sizes, but good none the less.

Submitted by Assigned_Reviewer_22

In "Learning the Learning Rate for Prediction with Expert Advice", the authors build on the adaptive version of Hedge to produce an algorithm that does as well as best learning rate and also gives second order worst case regret bounds for hedge. The body of the paper also contains a simulation/plot which shows how an intermediate learning rate hedge can perform much better than a worst case rate learning rate.

Hedge is simple algorithm which predicts an action in proportion to its exponentially weighted loss. Thus the base of the exponent is the learning rate. If one does not know either an upper bound on the loss of the best action or the number of trials there are a number of adaptive schemes that adapt the learning rate.

The paper is well-written but rather technical. Thus without more background it is difficult for me to assess the significance of the contribution. Thus to clarify I have the following for the authors regarding the significance and motivation for the bounds.

Regarding theorem 3.5, you say in the abstract that the regret is only increased by a polylogarithmic factor. Why is this "okay"? it seems even a constant multiplicative factor increase is a large price to pay in a worst-case setting. Can you argue why a constant or larger factor is necessary.

Regarding Figure 1 (Theorem 3.5) to a naive reader as myself, it would seem that Figure 1 is an intuitive motivation why we would be interested in bound like 3.5. Unfortunately, I do not have intuition prior to the appendix on the type of data that would generate that plot, if possible it would be nice to see that motivation returned to the paper. In theorem 3.5 the comparison class is implicitly a set of *adaptive* algorithms (and thus hard to understand their significance). Is it easy to factor out \eta and simply give a bound in terms of "sequence properties"?

In Theorem 3.6 how do these second-order bounds compare to those of [7] and also those in Extracting Certainty from Uncertainty: Regret Bounded by Variation in Costs (Hazan & Kale)?
Summary: Well-written technical contribution. Difficult for me to assess the significance of this paper without a further background in second order bounds or more illustrative examples in the paper.

(Updated to score qual. 6->7 based on response)
Author Feedback
Author rebuttal: Dear reviewers, Thank you for your comments and to-the-point questions.

Reviewer_19:

Q "1. I do have some concerns over the lower limit of learning rate used though [...] I wonder if one could get a guarantee over entire range of step sizes [...]."

A Equation 6 does give a guarantee for arbitrarily small eta in the form of a bound R_T = O(log(K)/eta). Admittedly, it is possible that this bound might exceed the actual regret of eta, but we still think this is a pretty good bound, as it improves the standard upper bound on the regret from Equation 1 (because delta_t^eta >= 0).

Q "2. [...] one might consider hindsight-comparison to step sizes of form eta_t = eta / t^a where eta and a are tuned. The result in this paper only refers to a = 0.[...] it is not clear to me that the results can be adapted to nonzero a."

A Indeed, we have been thinking about this exact same question ourselves! Our current best answer is this: we believe our technique can be directly applied to learn step sizes of form eta_t = eta / t^a for fixed eta and tunable a, giving an algorithm that has regret within polylog factor of the same fixed eta and the hindsight-optimal a. The crucial properties that these step sizes share with the grid of eta from the paper are that 1) they are nonincreasing; 2) different parameters never cross: if a1 < a2, then eta / t^{a1} > eta / t^{a2} for all t. If we also allow eta to be a parameter, then the second property might sometimes be violated, but so rarely that we suspect our algorithm can be easily adjusted to deal with this. In any case, the ability to handle time-varying step sizes where only 'a' is a parameter already seems promising.

Reviewer_22:

Q "[...] the regret is only increased by a polylogarithmic factor. Why is this "okay"? [...] Can you argue why a constant or larger factor is necessary."

A Just to be sure we are on the same page: the factor occurs in front of the *regret* for any fixed choice of eta; it does not multiply, say, the cumulative loss for eta or anything, which would definitely be unacceptable.

About the regret: for worst-case data, we know that the regret grows like O(T^{1/2}) and, for very benign data, we know that the regret of Hedge with optimally tuned learning rate can be as small as a constant = O(T^0). It therefore seems natural to interpolate by saying, for alpha in [0,1/2], "the data are alpha-easy" if the regret of Hedge with the hindsight-optimal \eta grows as O(T^alpha). So then the important thing is to get the exponent alpha right. As shown around line 116 of the paper, our multiplicative factor is at most O( ln^{1+epsilon}(T) ), i.e. polylogarithmic. This means that instead of T^alpha we get O(T^beta) for *every* beta > alpha, which we find acceptable, because it gets arbitrarily close to the smallest possible exponent alpha.

Q "[...] it would seem that Figure 1 is an intuitive motivation why we would be interested in bound like 3.5. Unfortunately, I do not have intuition [...] on the data that would generate that plot"

A Indeed, giving an intuitive motivation is exactly why we added Figure 1. At a high level, it is very simple: there exist some data for which large eta is much better than small eta, and there also exist data for which small eta is much better than large eta. We simply put these two types of data together, one after the other, which ensures that some intermediate eta will be the best, and then we repeated the same pattern many times. In practice, especially when the number of experts is large, there might be other, more complicated interactions between experts that cause intermediate eta to be the best, but our current approach seemed to us the easiest way to make our point.

Q "[...] the comparison class is [...] a set of *adaptive* algorithms (and thus hard to understand their significance). Is it easy to factor out \eta and simply give a bound in terms of "sequence properties"?"

A Regarding the significance of Hedge, being an adaptive algorithm:
- Different choices of eta cover important special cases: eta = infinity gives FTL; eta = 1 gives Bayesian probability updating if the expert losses are logarithmic losses; for eta = sqrt{8 ln(K)/T} Hedge achieves the worst-case bound.
- Hedge is the seminal algorithm for prediction with expert advice, and much prior work has been done on tuning eta

Regarding "factoring out eta": it is known that if losses are iid and/or the number of leader changes is finite,then for large enough n, eta = infty is optimal; and if certain Bayesian assumptions hold, then eta = 1 is optimal; but unfortunately, we know of no straightforward general mapping between easy-to-formulate sequence properties and corresponding optimal eta's. So factoring out eta does not seem easy.

Q "In Theorem 3.6 how do these second-order bounds compare to those of [7] and [...] (Hazan & Kale)?"

A The bounds in Thm 3.6 are of the general forms expressed in Equations 2 and 3, so we recover the previous results obtained by [7]. For the best known constants in this case we refer to [5].

The results of Hazan&Kale are very interesting, but since they modify Hedge with an extra quadratic loss term, they do not fall directly into the scope of our comparison class. They bound regret in terms of the cumulative variance of the losses of the best expert. If there is any noise in the data at all, then we would expect this cumulative variance to grow linearly with T, so their bound would be of the order Omega(T^(1/2)), which is the worst-case regime.

Reviewer_6:

Q "Are [...] the new parameters that the algorithm needs as input [...] robust enough so that we do not need to learn them, too."

A Tuning eta is much more difficult, because it can have a large impact on the regret, and we expect the optimal value for eta to change with T. By contrast, the new parameters only trade off the constants in the bounds, so their effect is not nearly as large as the effect of eta.